# Multiple Sclerosis: Melatonin, Orexin, and Ceramide Interact with Platelet Activation Coagulation Factors and Gut-Microbiome-Derived Butyrate in the Circadian Dysregulation of Mitochondria in Glia and Immune Cells

**DOI:** 10.3390/ijms20215500

**Published:** 2019-11-05

**Authors:** George Anderson, Moses Rodriguez, Russel J. Reiter

**Affiliations:** 1CRC Scotland & London, London E16 6JE, UK; 2Departments of Neurology & Immunology, Mayo Clinic College of Medicine & Science, Rochester, MN 55905, USA; rodriguez.moses@mayo.edu; 3Department Cellular and Structural Biology, Uni Texas Health Science Center, San Antonio, TX 78229, USA; REITER@uthscsa.edu

**Keywords:** multiple sclerosis, gut dysbiosis, gut permeability, immune-inflammation, mitochondria, melatonin, orexin, platelets, circadian, treatment

## Abstract

Recent data highlight the important roles of the gut microbiome, gut permeability, and alterations in mitochondria functioning in the pathophysiology of multiple sclerosis (MS). This article reviews such data, indicating two important aspects of alterations in the gut in the modulation of mitochondria: (1) Gut permeability increases toll-like receptor (TLR) activators, viz circulating lipopolysaccharide (LPS), and exosomal high-mobility group box (HMGB)1. LPS and HMGB1 increase inducible nitric oxide synthase and superoxide, leading to peroxynitrite-driven acidic sphingomyelinase and ceramide. Ceramide is a major driver of MS pathophysiology via its impacts on glia mitochondria functioning; (2) Gut dysbiosis lowers production of the short-chain fatty acid, butyrate. Butyrate is a significant positive regulator of mitochondrial function, as well as suppressing the levels and effects of ceramide. Ceramide acts to suppress the circadian optimizers of mitochondria functioning, viz daytime orexin and night-time melatonin. Orexin, melatonin, and butyrate increase mitochondria oxidative phosphorylation partly via the disinhibition of the pyruvate dehydrogenase complex, leading to an increase in acetyl-coenzyme A (CoA). Acetyl-CoA is a necessary co-substrate for activation of the mitochondria melatonergic pathway, allowing melatonin to optimize mitochondrial function. Data would indicate that gut-driven alterations in ceramide and mitochondrial function, particularly in glia and immune cells, underpin MS pathophysiology. Aryl hydrocarbon receptor (AhR) activators, such as stress-induced kynurenine and air pollutants, may interact with the mitochondrial melatonergic pathway via AhR-induced cytochrome P450 (CYP)1b1, which backward converts melatonin to N-acetylserotonin (NAS). The loss of mitochnodria melatonin coupled with increased NAS has implications for altered mitochondrial function in many cell types that are relevant to MS pathophysiology. NAS is increased in secondary progressive MS, indicating a role for changes in the mitochondria melatonergic pathway in the progression of MS symptomatology. This provides a framework for the integration of diverse bodies of data on MS pathophysiology, with a number of readily applicable treatment interventions, including the utilization of sodium butyrate.

## 1. Introduction

The role of circadian dysregulation in multiple sclerosis (MS) is indicated by the modulation of its course by circadian gene alleles [1], and the increased risk associated with single nucleotide polymorphisms (SNPs) in the circadian genes *Aryl Hydrocarbon Receptor Nuclear Translocator Like (ARNTL)* and Circadian Locomotor Output Cycles Kaput (CLOCK) [2]. The heightened MS risk linked with latitudes further from the equator has classically been seen as indicative of a role for decreased vitamin D in the etiology and course of MS [3,4,5,6]. However, such data may also be attributed to an irregularity in the circadian rhythm, as supported by an increased risk of MS in shift-workers [7]. As circadian genes drive oscillations in mitochondrial rate-limiting enzymes [8], circadian dysregulation can impact core aspects of cellular metabolism. Such data indicate an important role for circadian rhythm alterations in MS, including via the modulation of mitochondrial function.

Growing bodies of data highlight the role of the gut microbiome in the pathophysiology of a host of medical conditions [9,10], including MS [11]. Gut dysbiosis and increased gut permeability are associated with heightened levels of oxidative stress and immune-inflammatory activity, as well as raised levels of circulating lipopolysaccharide (LPS). This is coupled with a decrease in the gut microbiome-derived short-chain fatty acid, butyrate. Butyrate has a number of protective effects, including maintaining the gut barrier, suppressing immune responsivity, and optimizing mitochondrial function (reviewed in [9]). Many of the effects of stress, an MS etiological and relapse risk factor [12], are mediated via an increase in gut permeability/dysbiosis and associated decrease in butyrate release [13]. As the gut is also an intimate aspect of the circadian rhythm, gut alterations are linked with a wide array of MS pathophysiological data, including mitochondria, stress, immune cell responsivity, glia activity, and oxidative stress, as well as circadian dysregulation.

This article reviews data on the role of decreases in pineal gland-derived night-time melatonin and daytime orexin levels in the circadian dysregulation of mitochondria functioning, linking wider, previously disparate, bodies of data on the pathoetiology and pathophysiology of MS. It is proposed that gut dysbiosis and gut permeability increase circulating LPS and other factors and processes that activate microglia. Microglia activation increases the production and release of tumor necrosis factor (TNF)-α and peroxynitrite (ONOO^-^), with the latter elevating levels of astrocyte acidic sphingomyelinase (aSMase), in turn increasing ceramide release, including within exosomes. Ceramide is a major mediator of mitochondrial dysregulation across an array of different cell types. Both TNF-α and ceramide suppress levels of daytime orexin and night-time pineal gland-derived melatonin, thereby suppressing the daytime and night-time optimization of mitochondrial functioning and oxidative phosphorylation by orexin and melatonin, respectively. The loss of this circadian rhythm regulation of optimized mitochondrial function alters how mitochondria act to co-ordinate cellular function across different cell types, including oligodendrocytes, immune cells, and cells at the blood–brain barrier (BBB).

Consequently, gut dysregulation modulates MS pathophysiology via a number of routes: (1) LPS ultimately activates ceramide, which increases apoptotic susceptibility via detrimental impact on mitochondrial function; (2) ceramide and associated inflammatory cytokines suppress the wake promoting and sleep promoting effects of orexin and melatonin, respectively; (3) the suppression of orexin and melatonin disrupts the circadian rhythm, including from the loss of the mitochondria optimizing effects of orexin and melatonin; (4) the attenuation of gut butyrate production contributes to suboptimal mitochondrial function, which increases apoptotic susceptibility, as well as the reactivity of immune cells, glia, and platelets, contributing to a wider pro-inflammatory milieu. Such a model links the wide array of diverse bodies of data pertaining to MS, including data on stress, obesity, melatonin, orexin, the kynurenine pathway, oxidative stress, depression, inflammation, gut dysregulation, mitochondria dysfunction, platelet activation, thrombin, and increased myocardial infarction risk.

First, we review wider bodies of data on MS pathophysiology, before integrating these into a model that highlights the circadian regulation of mitochondrial function, particularly via the mitochondrial melatonergic pathway and how this may be intimately linked to alterations in the gut microbiome.

## 2. MS Pathophysiology

### 2.1. Circadian Rhythms, the Gut Microbiome, and Melatonin

#### 2.1.1. Circadian Rhythms

Low vitamin D and increased body mass index (BMI) can heighten the risk of pediatric MS [14]. Notably, both low vitamin D and increased obesity/high BMI are associated with gut dysbiosis and raised levels of gut permeability [15], with gut dysbiosis linked to circadian dysregulation [16]. This supports the hypothesis that some of the seemingly small effects of vitamin D in MS [17,18] may be additive with other factors, in the regulation of the gut microbiome, with consequences for circadian rhythm regulation.

Alterations in circadian genes modulate the gut microbiome [19], whilst changes in the gut microbiome significantly change the circadian rhythm, as shown in a murine model [16]. Consequently, drivers and indicants of circadian dysregulation, including sleep loss and shift-work, are proposed to have their pathophysiological influence on metabolic disorders via mechanisms that are partly driven by alterations in the gut microbiome [20]. This provides a wider context for the data showing that shift-workers have an increased risk of MS [7]. A preclinical model of shift-work (inverted day–night every two weeks for eight weeks) showed a shift in the gut microbiome that increased gut permeability, coupled with alterations in intestinal epithelial cell exosome content, circulating LPS, and increased M1-type pro-inflammatory macrophages [21]. A reduction in the gut microbiome-derived short-chain fatty acid, butyrate, also contributes to the association of gut dysbiosis with metabolic functioning [22]. Overall, changes in the circadian rhythm may be intimately intertwined with alterations in the gut, and thereby to a host of pathophysiological processes.

#### 2.1.2. Gut Microbiome-Derived Butyrate and Mitochondria

Butyrate maintains the gut barrier, thereby preventing a “leaky gut” and the associated increases in oxidative stress, immune–inflammatory activity, and circulating LPS, which all compromise mitochondria, and therefore metabolic function [9]. An “in vitro oligodendrocyte + microglia” model of MS showed the effects of LPS in decreasing oligodendrocyte myelin production via microglia activation, which increased inducible nitric oxide synthase (iNOS) and nicotinamide adenine dinucleotide phosphate oxidase (NADPH oxidase), leading to an increase in oxidative and nitrosative stress (O&NS), especially via peroxynitrite (ONOO^-^) [23]. The elevated ONOO^-^ and O&NS detrimental effects in oligodendrocytes are negated by optimizing mitochondria oxidant status and function [23]. As such, alterations in the gut microbiome may have an impact on glia interactions that leads to suboptimal mitochondrial functioning in microglia and oligodendrocytes, which upregulates oxidants and decreases their capacity to deal with oxidant challenge, thereby suppressing myelin synthesis. Butyrate can suppress the effects of LPS across a number of cell types [24,25,26], including microglia [27], suggesting that a drop in circulating butyrate will compromise the ability of glia to respond to challenges, including LPS. Some efficacy of butyrate seems to be mediated by its induction of the melatonergic pathway, as shown in intestinal epithelial cells [28]. This suggests that the loss of gut-derived circulating butyrate in MS will potentiate the LPS effects in microglia, thereby enhancing the suppression of oligodendrocyte mitochondrial function, including via a putative decrease in the activation of the mitochondrial melatonergic pathway. Consequently, there is an attenuation of the ability of mitochondria to respond to ONOO^-^ and O&NS challenge [9]. Such oligodendrocyte dysfunction leads to the concept of “dying-back oligodendrogliopathy” [29].

Circulating butyrate has positive effects on a number of cell types, including immune- and glia-dampening effects [30]. Many of these effects are mediated by its histone deacetylase (HDAC) inhibition and the optimization of mitochondrial function [9]. Consequently, the associations of gut dysbiosis and decreased butyrate with metabolic and circadian dysregulation in MS may arise from suboptimal mitochondrial function [9]. Such processes are also relevant to a wide array of other medical conditions [31,32,33,34]. In vitro and preclinical data showed butyrate to inhibit demyelination and enhance remyelination, via increasing the differentiation of immature oligodendrocytes [35], which the authors propose are via the direct HDAC inhibitory effects on oligodendrocytes. Oligodendrocyte precursor cells (OPCs) at MS lesions are proposed to interact with endothelial cells and astrocyte end-feet, leading to an increase in BBB permeability, thereby contributing to immune cell extravasation and MS symptomatology [36]. This suggests that the suppression of OPC differentiation and increased BBB permeability may be coordinated by astrocytes. A number of other factors act to inhibit remyelination, including immune, miRNAs, and aging [37], with the mitochondrial optimizing effects of creatine increasing oligodendrocyte survival and OPC differentiation [38]. Such data highlight the importance of gut microbiome-derived butyrate, via optimized mitochondrial function, for the regulation of classical MS pathophysiology.

A growing body of evidence shows alterations in mitochondria functioning in MS, including in CNS cells and immune cells. Variations in immune cell mitochondrial function can dramatically alter the nature and pattern of immune responses, as evidenced in primary versus secondary progressive MS T-cells, which show differential responses that are mediated by mitochondrial alterations [39]. Butyrate may have direct and indirect exosome-driven effects on mitochondria and cellular functioning, including via its induction of the mitochondrial melatonergic pathway in systemic and CNS cells [9,28].

#### 2.1.3. Pineal Gland and Mitochondria Melatonin

Melatonin is classically associated with its night-time production by the pineal gland, from where it is important in driving and regulating the circadian rhythm. However, data show melatonin to be produced by all mitochondria-containing cells. In fact, mitochondria contain all the enzymes and co-factors necessary for melatonergic pathway activation, including aralkylamine *N*-acetyltransferase (AANAT), *N*-acetylserotonin o-methyltransferase (ASMT), acetyl-CoA, and 14–3-3 proteins. Melatonin has a number of effects that optimize mitochondrial function, including antioxidant, membrane structure, and sirtuin-3 induction, with the latter leading to an increase in manganese superoxide dismutase (SOD2), thereby increasing endogenous antioxidant potential [40,41,42]. It requires investigation as to whether butyrate increases the activation of the mitochondrial melatonergic pathways, and thereby mitochondria and cellular functioning, across different cell types, including immune and CNS cells.

#### 2.1.4. Mitochondria at the Heart of Circadian Rhythm

Recent work on pineal gland-derived circadian melatonin indicated that it may mediate many of its effects via the upregulation of the mitochondrial melatonergic pathway [41]. It is now appreciated that all cells show variations in glycolysis and oxidative phosphorylation, with the former classically associated with cancer cells (Warburg effect). Reiter and colleagues proposed that daytime stressors/toxins/epigenetic processes may increase daytime glycolysis levels in cancer cells and other cell types, which can become reset to oxidative phosphorylation by adequate levels of pineal gland-derived night-time melatonin [41,42]. These authors suggest that this is mediated by the uptake of pineal gland-derived melatonin into mitochondria, via peptide transporters (PEPT)1/2, with melatonin stimulating activation of the mitochondrial melatonergic pathway. This is proposed to be achieved via melatonin inhibiting pyruvate dehydrogenase kinase (PDK), in turn disinhibiting the pyruvate dehydrogenase complex (PDC) and thereby increasing production of acetyl-CoA from pyruvate. Acetyl-CoA is a necessary co-factor for AANAT and initiation of the mitochondrial melatonergic pathway (see Figure 1). As such, pineal gland-derived melatonin may act to reverse some daytime challenges that drive shifts in mitochondrial functioning and better optimize, if not reset, mitochondria metabolism, at least in part via regulation of the mitochondrial melatonergic pathway. Such a perspective puts mitochondria at the heart of the circadian rhythm.

### 2.2. Ceramide, Mitochondria, and MS

#### 2.2.1. Ceramide, Oxidants, and Acid Sphingomyelinase

Cerebrospinal fluid (CSF) from MS patients significantly alters neuronal mitochondria shape and glucose uptake, coupled with a suppression of mitochondria complexes I, III, and IV, as well as reduced ATP production. The authors attribute this to CSF exosomes containing ceramides and the phosphorylation of dynamin-related protein (DRP)1 [43]. Such data link to previous studies showing suboptimal mitochondria functioning in MS and indicate an attempt to increase glycolysis under challenge [44]. Such findings also highlight the growing body of data showing the importance of ceramide in MS pathophysiology [45]. A corollary of ceramide-induced alterations in mitochondrial metabolic functioning is a decrease in pyruvate uptake, leading to an increase in its extracellular conversion to lactate, with higher CSF lactate levels evident in MS, in correlation with symptom severity [46]. Metabolic alterations are also evident in MS T-cells [47], suggesting wider metabolic dysregulation.

As indicated above, the rise in gut dysbiosis/permeability elevates circulating LPS levels as well as intestinal exosomes containing high-mobility group box (HMGB)1 [48]. HMGB1 is increased in systemic and CNS cells in MS [49]. Via the activation of toll-like receptors (TLR), both LPS and HMGB1 elevate iNOS and superoxide, leading to ONOO^-^ formation and an increase in pro-inflammatory cytokines, including TNF-α. Notably, TNF-α, iNOS, and NADPH oxidase are all increased in MS [50]. Although iNOS is increased in macrophages, activated microglia, astrocytes, and oligodendrocytes surrounding MS lesions, it is its conversion to ONOO^-^ that is damaging to cells in MS [51], including oligodendrocytes [52]. Notably, ONOO^-^ is a particularly powerful inducer of aSMase and thereby ceramide synthesis. Astrocytes may be a particular target for LPS/HMGB1 triggered ONOO^-^ and its induction of aSMase and ceramide. Ceramide-regulated CNS glia and their interactions are significant treatment targets in MS [53].

#### 2.2.2. Astrocytes and Ceramide

Astrocytes play a major role in the co-ordination of brain function. Astrocytes significantly regulate neuronal activity and are key modulators of axonal activity, particularly via ionic regulation at the nodes of Ranvier. Astrocytic end-feet are a key aspect of the structure and function of the BBB. By forming gap junctions, astrocytes are intimately integrated into astrocytic networks, allowing for wider coordinated neuronal regulation. As such, astrocytes are key structural and functional aspects of the two key sites classically associated with central MS pathophysiology, viz the BBB and axonal nodes of Ranvier. Perinodal astrocytes can reversibly alter myelin thickness, conduction velocity, and nodal gap length [54]. These authors showed that thrombin-induced detachment of myelin from the axon is inhibited by the release of astrocyte vesicle associated membrane protein (VAMP)2, a thrombin protease inhibitor [54], highlighting the powerful role of astrocytes in white matter regulation. Thrombin and fibrinogen, two predominantly blood-derived products, have long been associated with MS, primarily via effects on demyelination and the inhibition of OPCs differentiation [55]. The data of Dutta and colleagues indicate that perinodal astrocytes have a powerful role in the regulation of the effects of thrombin on myelin and axonal dysregulation [54]. The role of coagulation factors in MS will be elaborated on below. Suffice to note that astrocyte responses can be a determinant of the damaging effects of such factors.

As well as direct effects on astrocytes, gut-derived LPS/HMGB1, and other TLR2/4 factors, activate microglia to increase iNOS and superoxide, leading to ONOO^-^. ONOO^-^ stimulates astrocyte aSMase, leading to the production of longer chain ceramides. As well as altering astrocyte mitochondria and thereby wider cellular and intercellular functioning, the induction of ceramide has other consequences. Under challenge, exosomes from astrocytes can contain longer chain ceramides, which, at sufficient levels, can be pro-apoptotic [56]. A rise in astrocyte ceramide may be a significant indicator, and driver, of central inflammatory processes across a diverse array of distinct neurodegenerative conditions [57]. Challenged astrocytes are an important source of ceramide, with negative consequences at the BBB as well as in oligodendrocytes. The genetic inhibition of aSMase/ceramide prevents the classical MS-like pathophysiology in the experimental autoimmune encephalomyelitis (EAE) model, including BBB disruption, leukocyte extravasation, and demyelination [58]. The efficacy of fingolimod in the treatment of MS is classically attributed to its inhibition of thymic leukocyte egress. However, fingolimod also attenuates ceramide-induced BBB dysfunction in MS via effects on reactive astrocytes [59]. In active MS lesions there is an increase in ceramide synthesis, via raised astrocyte aSMase. Elevation in TNF-α and ONOO^-^ also increase astrocyte aSMase levels [59]. Thus, astrocyte-derived ceramide at key sites and in key processes is linked to MS pathophysiology, including via the release of ceramide in exosomes.

As well as detrimental effects at the BBB and in white matter, ceramide can impact immune cell functioning. In microglia, ceramide increases assembly of the NOD-, LRR-, and pyrin domain-containing protein (NLRP)3 inflammasome, thereby increasing the levels and release of the pro-inflammatory cytokines interleukin (IL)-1β and IL-18 [60]. This further contributes to the inflammatory milieu. TLR2-dependent ceramide induction in macrophages leads to suppression of mitochondrial respiration and the redirecting of the tricarboxylic acid metabolites to glutathione synthesis and inflammatory gene induction [61]. Effects of ceramide are mediated via conversion of macrophages to an M3-like non-aerobic glycolysis phenotype. The importance of mitochondria to immune cell phenotypes is also highlighted by the data showing that the M2-like anti-inflammatory, phagocytic macrophage phenotype uses oxidative phosphorylation [62], whilst the M1-like proinflammatory phenotype uses glycolysis [63]. In astrocytes, ceramide prevents the transport of mitochondria ADP/ATP, highlighting the role of ceramide effects in mitochondria that modulate the activity of reactive cells [64]. Ceramide is an important regulator of mitochondria physiology and therefore to the responses of immune and glia cells. Clearly, an increased release of longer chain ceramides from astrocytes will have wide-ranging impacts on many cell types and systems, at least in part via the regulation of mitochondrial function.

Human monoclonal antibodies and human immunoglobulin (Ig)Ms can promote remyelination and significantly raise the levels of myelinated axons in MS preclinical models [65], with effects that seem driven by decreasing astrocyte aSMase and increasing the S1P/ceramide ratio [66]. These natural human monoclonal antibodies have been found to be safe in over 70 patients with severe deficits from MS [67].

#### 2.2.3. Oligodendrocytes and Ceramide

Ceramide can lead to oligodendrocyte apoptosis, and potentiate pro-inflammatory cytokine effects on such apoptosis, whilst cytokines can increase ceramide release in exosomes [68]. Consequently, suppressing ceramide synthesis has been proposed as a new treatment target in MS [69]. The increased myelin regeneration by aSMase inhibition is attributed to the ensuing drop in ceramide levels [70]. Such negative effects of aSMase and ceramide are mediated via alterations in mitochondria functioning [71], with effects regulated by variations in mitochondrial sirtuin-3.

The detrimental effects of ceramide include its induction of mitochondria dysfunction. The raised levels of ceramide over aging correlate with senescence-associated changes in cells, which seem primarily driven by changes in mitochondrial function [72]. Ceramide effects on mitochondria have focused on its pro-apoptotic properties, clearly relevant to the loss of oligodendrocytes and myelination in MS. However, some of the effects of ceramides can be more subtle, with ceramide driving changes in the ratio of glycolysis/oxidative phosphorylation [73]. Ceramide also regulates other factors that modulate mitochondrial function, including suppressing levels of 14–3-3 and thereby decreasing the stabilization of AANAT, the initial enzyme in the mitochondrial melatonergic pathway. A decrease in AANAT stabilization means the loss of mitochondria melatonin, including loss of its positive effects on oxidative phosphorylation, sirtuin-3, and antioxidant regulation [9,40]. Ceramide may also suppress orexin, the hypothalamic wake-promoting factor [74], which has mitochondrial and melatonergic pathway regulatory effects, as detailed below.

#### 2.2.4. Butyrate and Ceramide

Increasing gut microbiome-derived butyrate lowers circulating levels of ceramides and sphingosines in a preclinical model [75], indicating a role for butyrate in suppressing the detrimental effects of ceramide in the mitochondria of many cell types in MS. As noted, the suppression of ceramide is a significant treatment target in MS [69,70], suggesting that the putative direct protective effects of butyrate in oligodendrocytes and astrocytes may be linked to the suppression of ceramide in these cells. The loss of butyrate is strongly associated with an increase in gut permeability, and therefore with heightened levels of LPS, HMGB1, oxidants, and immune inflammatory activity. Under such circumstances, the loss of butyrate’s suppression of ceramide and the loss of butyrate’s optimization of mitochondrial function may have additive effects that contribute to MS pathophysiology. Obesity and a high fat diet can increase gut dysbiosis/permeability, as well as raising circulating levels of ceramides [76]. Obesity increases MS risk as well as contributing to a more severe clinical course [77]. The BMI in MS modulates ceramide-induced DNA methylation and disease course, via increased levels of monocytes [78]. Gut-derived butyrate is significantly decreased in obesity, and is an important driver of the alterations in the gut–liver axis that underpin obesity, non-alcoholic fatty liver disease, and wider metabolic dysregulation [79], all of which may be suppressed by fingolimod, suggesting unrecognized gut microbiome, butyrate/HDAC inhibitory-like effects of this drug [80,81].

Butyrate also raises levels of glucosylceramide synthase [82], which adds a glucosyl moiety to ceramide, being the first step in the production of more complex glycosphingolipids, such as lactosylceramide and gangliosides [83]. This is one means by which butyrate may decrease levels of circulating ceramides. This is important, given that gangliosides increase OPC differentiation, with gangliosides being a significant MS treatment target [84]. It is unknown whether the efficacy of butyrate in OPC differentiation is mediated, at least partly, via increasing glucosylceramide synthase and ganglioside production. Fingolimod, like butyrate, is an HDAC inhibitor and increases gangliosides [85]. It is also of note that the gut microbiome exhibits a circadian rhythm [86], as does ceramide [87].

### 2.3. Orexin, Melatonin, and Mitochondria Circadian Regulation

Orexin and melatonin are two factors that are often decreased in MS and have some utility in its treatment. Although having opposing effects on arousal, both factors are associated with the optimization of mitochondrial function. Both orexin and melatonin increase the activity of the mitochondrial melatonergic pathway, thereby driving an increase in oxidative phosphorylation. Ceramide and TNF-α suppress orexin and pineal gland-derived melatonin, thereby suppressing the circadian influence of these factors on mitochondrial function (see Figure 1). TNF-α seems to have complex effects on MS [88], both detrimental and beneficial, likely as a consequence of dysregulated inflammatory and counter-regulatory anti-inflammatory responses [89].

#### 2.3.1. Orexin: Wake Promotion and Mitochondria

Orexins have important roles in the regulation of sleep and arousal states, as well as appetite, food intake, and energy homeostasis [90]. Orexin antagonists can also attenuate depression severity, at least in part via their impact on the sleep–wake circadian cycle [91]. A decrease in serum orexin-A level is evident in MS, where it correlates with symptom severity as well as with reductions in serum brain-derived neurotrophic factor (BDNF) and melatonin [92]. The peripheral administration of orexin A ameliorates EAE symptoms, primarily by decreasing CNS neuroinflammation, and, to a lesser extent, by suppressing demyelination or lowering astrocyte and microglia reactivity [93].

In the EAE model, orexin-A administration has positive effects [94], not only attenuating clinical symptoms, but also suppressing leukocyte infiltration, whilst up-regulating myelin basic protein and transforming growth factor (TGF)-β mRNA and down-regulating iNOS, matrix metalloproteinase (MMP)-9, and IL-12 mRNA. The orexin-1 receptor antagonist reversed these effects [94]. As hypothalamic lesions are not uncommon in MS [95], including as initial lesion sites, suppressed orexin levels may be a consequence. Low orexin levels are associated with excessive daytime sleepiness [96,97], fatigue, and functional disability in relapse-remitting MS (RRMS) [98], as well as with depression and anxiety [99]. Fatigue is also associated with suboptimal mitochondria functioning, with dietary attempts to regulate fatigue in MS being modeled on mitochondrial regulation [100]. The association of orexin with mitochondrial function may be of some importance in MS pathophysiology [101]. MS patients with a history of optic neuritis also show increased rates of daytime sleepiness in association with lower levels of both orexin and melatonin [102], suggesting significant impacts on the circadian regulation of mitochondria functioning.

Orexin levels and neurons are lost over ageing and contribute to neurodegeneration and neuroinflammation, which may be exacerbated by obesity and a high-fat diet [103]. Orexin leads to the disinhibition of PDC, thereby increasing the conversion of pyruvate to acetyl-CoA, contributing to increased activation of the melatonergic pathway and paralleling the effects of pineal gland-derived melatonin [41,42], as shown in Figure 1. It requires investigation in CNS cells as to the role of orexins in the regulation of the circadian shifts in mitochondrial function, including whether these are mediated via the upregulation of the mitochondrial melatonergic pathway.

Over 50% of MS patients show insulin resistance, which becomes more prevalent in secondary progressive MS (SPMS) [104]. Of note, both orexin and melatonin are decreased in SPMS [92], with both of these factors associated with the inhibition of insulin resistance [79], as well as the maintenance of mitochondrial oxidative phosphorylation [105]. Preclinical data indicate orexin to be important in the regulation of insulin resistance to an array of different challenges, including social defeat stress [106]. Orexin effects on the autonomic balance can bidirectionally regulate hepatic gluconeogenesis, thereby generating oscillations in the daily blood glucose levels. The decrease in orexin levels over the course of aging is associated with heightened levels of endoplasmic reticulum (ER) stress [107], with ER stress associated with alterations in mitochondrial functioning, including during the course of insulin resistance [108].

#### 2.3.2. Orexin and Obesity in MS

At diagnosis, alterations in lipid profiles and obesity are coupled with enhanced levels of central inflammation and heightened clinical disability in RRMS. It is thought that the raised levels of adipocytokines and lipids may underpin the negative impact of increased BMI on the course of RRMS [77]. Elevated BMI also raises the risk of pediatric MS, as well as contributing to a poorer treatment response in pediatric MS [109], although the associations of pediatric MS and obesity may be partly mediated by an earlier puberty [110]. As noted, increased BMI heightens ceramide levels [76], contributing to ceramide-induced DNA methylation and disease course, including demyelination and raised monocyte levels [78]. Heightened ceramide levels also contribute to obesity via increasing lipotoxicity, oxidative stress, and ER stress in the ventromedial hypothalamus (VMH) [111], thereby driving obesity in the absence of hyperphagia [112]. Alterations in VMH activity act to regulate orexin neuronal activation [113], with orexin neurons projecting to the raphe pallidus, leading to heightened levels of sympathetic activation of brown adipose tissue (BAT) [114]. This orexinergic pathway may also drive the browning of white adipose tissue (WAT) [115]. Orexin neuronal projections to the rostral ventrolateral medulla drive sympathetic splanchnic nerve activation and the release of epinephrine from the adrenal medulla, thereby contributing to orexin’s arousal associated effects [116]. Such effects of orexin highlight its importance in the association of obesity and arousal/fatigue with MS pathophysiology.

Orexin neurons can also be activated by reward cues as well as by reward per se, allowing orexin to be associated with the modulation of motivated behaviors [117]. In this context, orexin neurons project to the ‘reward-associated’ dopamine neurons of the ventral tegmental area (VTA), with orexin in the VTA raising the levels of both *N*-methyl-D-aspartate (NMDA) and α-amino-3-hydroxy-5-methyl-4-isoxazolepropionic acid (AMPA) receptors. Interestingly, as well as often co-releasing dynorphin, orexin neurons can release glutamate, which is proposed to potentiate VTA-associated reward signaling [118]. Overall, the lower orexin levels in MS will have a number of wider implications in regard to glucose monitoring and regulation, as well as in obesity susceptibility and alterations in reward processing.

#### 2.3.3. Orexin and Dynorphin

A number of other factors can be co-released with orexin, including the κ-opioid receptor agonist dynorphin, which is important for the regulation of rapid eye movement (REM) sleep [119]. Dynorphin activation of the κ-opioid receptor increases remyelination, suggesting its potential utility in MS management [120]. Unfortunately, systemically targeting an increased κ-opioid receptor activation can result in dramatic increases in dysphoria, driven by the increased activation of the amygdala κ/µ-opioid receptor ratio [121]. However, the relevance of factors, such as dynorphin, that are co-released with orexin requires investigation, including as to how these could act to modulate orexin effects on mitochondrial metabolic activity. For example, in cardiomyocytes, dynorphin can have protective effects that are driven by its opening of the mitochondria ATP-sensitive K^+^ channel (KATP) [122]. It may be important to investigate if mitochondria KATP channel openers, such as diazoxide, have protective effects in the EAE model [123], which would suggest that the co-release of dynorphin from orexin neurons may have mitochondria regulatory effects. It is also of note that dynorphin, like other opioid receptor agonists, acts on orexin neurons to increase orexin synthesis [124], indicating that some of the benefits of κ-opioid receptor agonism in the upregulation of remyelination may be mediated by increased orexin. The effects and consequences of the orexin co-released factors, including dynorphin and glutamate, will be important to determine.

#### 2.3.4. Orexin, Stress, and the Kynurenine Pathway

Orexin can also modulate the stress response, with variations in orexin contributing to stress resilience, susceptibility, and stress-induced depression [125], whilst sex differences in stress effects on orexin may contribute to the sex differences evident in MS [126]. Stress contributes to the etiology and relapse in MS [127], with stress, and associated depression, long appreciated to be predictive of relapse in RRMS. However, the biological underpinnings of stress/depression interactions with MS etiology and course are still to be determined. Stress, via an increase in corticotrophin releasing hormone (CRH), can elevate gut permeability, thereby raising levels of O&NS and pro-inflammatory cytokines, in turn activating indoleamine 2,3-dioxygenase (IDO) and tryptophan 2,3-dioxygenase (TDO), and consequently activating the kynurenine pathway, whilst also driving down levels of serotonin, *N*-acetylserotonin (NAS), and melatonin [128,129]. As such, stress effects in the gut may be associated with circadian dysregulation and alterations in mitochondria metabolism, partly via decreased levels of pineal gland- and mitochondria-derived melatonin [9,11,23]. In the cuprizone-induced demyelination model of MS, psychological stress potentiates levels of myelin degradation [130]. The loss of orexin neurons and orexin release in MS may be relevant to this, with preclinical data showing that orexin-A inhibition decreases serotonin neurons across different brain regions [131], indicating that the lower orexin-A levels in MS will decrease serotonin availability for the melatonergic pathway.

#### 2.3.5. Orexin, Melatonin, and Glutamate in MS

Pineal gland-derived melatonin has negative reciprocal interactions with lateral hypothalamic orexin neurons in preclinical models, suggesting that their interactions are important to circadian transitions in activity versus sleep [132,133]. In pinealectomized rodents, melatonin restoration suppresses the raised orexin levels induced by pinealectomy [134]. In a preliminary study of young men, Mäkelä and colleagues found a considerable variability in the levels and circadian fluctuations of orexin-A [135], indicating that orexin circadian variations require more extensive investigation, including their putative negative reciprocal interactions with pineal gland-derived melatonin.

The anti-aging effects of sirtuin-1 are proposed to be mediated by an increase in levels and activity of orexin-2 receptor in the dorsomedial and lateral hypothalamus [136]. This has benefits for mitochondrial function and sleep quality, key aspects in the pathoetiology of an array of neurodegenerative conditions, including MS. Given orexin increases arousal, its association with energy regulation would be expected. Hypothalamic orexin regulates endogenous glucose production via hypothalamic orexin output to the autonomic nervous system, thereby regulating hepatic glucoregulatory enzymes [137]. Although the aryl hydrocarbon receptor (AhR) is expressed in hypothalamic orexin neurons [138], the relevance of the AhR in these cells is unknown, including the AhR induction of cytochrome P450 (CYP)1b1. CYP1b1 metabolizes estrogen and can “backward” convert melatonin to NAS, with AhR-induced CYP1b1 thereby modulating mitochondrial function. Consequently, stress- and inflammation-induced TDO and IDO, via kynurenine and kynurenic acid mediated AhR-induced CYP1b1 [139], can interact with the estrogen and melatonergic pathways in orexin neurons, with impacts on mitochondrial physiology.

Increased glutamatergic activity contributes to neurodegeneration in MS, including when arising from raised IL-17 levels that suppress astrocyte glutamate transporters, GLT-1 and GLAST, as well as decreasing the conversion of astrocyte glutamate to glutamine [140]. Melatonin, via effects on mitophagy and the maintenance of mitochondrial function, affords protection in neurons challenged by raised glutamate levels [141]. Protection against glutamatergic toxicity is also afforded by orexin, which increases levels of astrocyte GLT-1 [142], highlighting another similarity in the effects of orexin and melatonin on MS pathophysiology.

## 3. Integrating MS Pathophysiology

The above data indicate a wide-acting role for gut dysbiosis/permeability in the modulation of a diverse array of data linked to MS pathophysiology. Gut dysbiosis/permeability increases oxidative stress, immune–inflammatory activity, circulating LPS, and exosomal HMGB1, which act on microglia to increase iNOS and superoxide, leading to ONOO^-^ induced aSMase and ceramides in astrocytes. Perinodal astrocyte-derived ceramide contributes to demyelination and suppressed remyelination, whilst also acting to increase BBB permeability and associated leukocyte extravasation. Ceramide alters mitochondria metabolic functioning, which may underpin its effects on oligodendrocytes and the BBB, with suboptimal mitochondrial function in axons long associated with MS [143]. Ceramide and TNF-α suppress both orexin and melatonin, thereby negatively regulating their putative daytime and night-time optimization of mitochondrial function. The efficacy of orexin and/or melatonin in the treatment of MS [144] indicate that the ceramide suppression of these two factors is relevant to the pathophysiology of MS. The benefits of orexin and melatonin on mitochondrial physiology may be mediated via their induction or potentiation of the mitochondria melatonergic pathway, driven by PDC disinhibition and raising acetyl-CoA levels. Melatonin increases mitochondria oxidative phosphorylation [41,145]. Ceramide, by decreasing 14–3-3, prevents the stabilization of AANAT and therefore melatonergic pathway activation [146]. Overall, gut dysbiosis/permeability, by increasing ceramide, has direct and indirect effects on mitochondrial function that underpin changes evident in MS. The suppression of orexin and melatonin mediates these indirect effects, and thereby contributes to the circadian dysregulation evident in MS.

The suppression of butyrate attenuates butyrate’s conversion of ceramide to gangliosides [83], thereby heightening the detrimental effects of ceramide and suppressing the protection afforded by some gangliosides, including ganglioside-induced OPC differentiation [84] (see Figure 1). Butyrate also disinhibits PDC [147], suggesting that butyrate may mediate its beneficial effects via upregulation of acetyl-CoA and the mitochondrial melatonergic pathway. As such, the gut microbiome may be intimately associated with the modulation of mitochondrial function, including mitochondrial circadian modulation, with consequences for the functioning of an array of diverse cell types, including glia and immune cell reactivity levels.

The diverse data on MS pathophysiology may be integrated within this simple framework, including the effects of MS susceptibility and relapse risk factors, such as stress/depression and obesity. Both stress/depression and obesity/high-fat–sucrose diets increase gut permeability, gut dysbiosis, circulating LPS, exosomal HMGB1, and levels of pro-inflammatory cytokines. All may contribute to MS pathophysiology. Pro-inflammatory cytokines also increase IDO and TDO activation, leading to an increase in kynurenines, thereby depleting levels of tryptophan availability for serotonin and melatonin synthesis. Kynurenine and kynurenic acid may also activate the AhR, leading to CYP1b1 induction, thereby increasing the metabolism of estrogen and driving the backward conversion of melatonin to NAS. As suppressed butyrate production raises ceramide levels and lowers levels of ceramide conversion to gangliosides, thereby suppressing orexin and contributing to suboptimal mitochondria functioning, the gut has emerged as an important hub for modulating and synchronizing many of the changes associated with MS pathophysiology.

Other data on MS pathophysiology may also be framed within this model, including the role of ceramide and gut dysbiosis in glucose dysregulation, insulin resistance, and the lipid alterations that are common in MS [148], with ceramide decreasing high-density lipoprotein (HDL) cholesterol and increasing very low-density lipoprotein (vLDL) cholesterol [149]. This model also provides a framework for incorporating the effects of medication in the treatment of MS, e.g., fingolimod inhibits aSMase and ceramide [80], including ceramide-induced emergent obesity in MS [150], and, like butyrate, increases gangliosides via HDAC inhibitory effects [85]. Although they have optimizing effects in mitochondria, orexin and melatonin can have distinct and sometimes opposing effects on glia and immune cells [151,152], which seems to contribute to the circadian variation in glia and immune cell activity [153].

Preclinical data show that high-fat diet-induced obesity results in the loss of the circadian rhythm of reactive cells [154], indicating that gut dysbiosis and ceramide-driven obesity, orexin/melatonin suppression, and mitochondria dysfunction will change the circadian regulation of reactive cell activity and function. As the immune system has a determining role on the survival and functioning of other body cells and systems, such changes in glia/immune cell functioning are crucial aspects of the dynamic alteration underpinning the pathoetiology and pathophysiology of MS. The gut microbiome influence on ceramide therefore mediates changes in mitochondria regulation and functioning in glia and immune cells. Clarification as to how dietary changes drive shifts in gut dysbiosis, thereby changing the nature of MS pathophysiology, will be important to determine.

The activation of the melatonergic pathway does not always lead to melatonin production. A number of factors can drive the ‘backward’ conversion of melatonin to NAS, including CYP1b1, CYP2C19, O-demethylation, high ATP, and metabotropic glutamate receptor subtype 5 (mGluR5) activation, (reviewed in [31]). NAS has many similar effects as melatonin, including antioxidant effects. However, NAS is a BDNF mimic, via its activation of BDNF receptor TrkB [155]. As such, the backward conversion of melatonin to NAS means the loss of melatonin effects in mitochondria, but also altered effects arising from raised NAS levels. NAS is increased in SPMS [156] in association with rises in kynurenine and kynurenic acid, indicative of these two kynurenines having activated the AhR to increase CYP1b1, thereby triggering the backward conversion of melatonin to NAS, coupled with an increase in the metabolism of estrogen. CYP1b1 SNPs are MS risk factors [157], highlighting the importance of CYP1b1 regulation in MS pathophysiology. The role of NAS in the changing nature of MS pathophysiology needs to be clarified, including whether NAS, like BDNF and other neurotrophins, activates p75NTR, which induces ceramide synthesis and is increased in glia of MS plaques [158]. This could suggest an important role for an increase in the NAS/melatonin ratio in the transition from RRMS to SPMS. As most of the increased NAS in the CSF of SPMM patients is likely derived from the pineal gland, it requires investigation as to the role of the pineal gland NAS/melatonin ratio, as well as any differential effects of NAS, versus melatonin, on mitochondrial function and mitochondrial melatonergic pathway activity across cell types.

Gut dysbiosis may also be associated with wider systemic changes altered in MS, and which also have relevance to classical MS central pathophysiology, including via the regulation of platelets and coagulation factors, such as thrombin and fibrinogen.

### 3.1. Platelets, Coagulation Factors, and Glia

Astrocytes are densely expressed along axons, especially at the nodes of Ranvier, where they are classically associated with the regulation of ionic fluxes at these demyelinated nodes. Axonal impulse transmission velocity is modulated by the thickness of the myelin sheath and nodes of Ranvier morphology. Astrocytes act to regulate the myelin thickness as well as nodal gap length, including protecting against thrombin-induced oligodendrocyte damage via the vesicular release of a thrombin protease inhibitor. Perinodal astrocytes, and factors acting on their function, are therefore important aspects of myelin regulation and of changes relevant to the early pathoetiology of MS [54]. Pro-coagulant factors, such as thrombin, can act on astrocytes to decrease glutamate transporters, and thereby increase glutamatergic damage [159]. Thrombin and fibrinogen, two predominantly blood-derived products, have long been associated with MS, predominantly via their effects on demyelination and the inhibition of OPC differentiation [55]. Such data also highlight the role of perivascular coagulation factors, including thrombin and fibrinogen, in central MS pathophysiology, primarily as a consequence of increased BBB permeability, and the compromised ability of perinodal astrocytes to negate such effects [54]. Gut-driven changes in ONOO^-^/aSMase/ceramide that impact astrocytic functioning therefore attenuate the ability of astrocytes to suppress the effects of thrombin and fibrinogen in oligodendrocytes and OPCs.

In progressive MS autopsies, there is evidence of extensive fibrin(ogen) deposition in correlation with a significantly lower neuronal density [160]. The effects of fibrin(ogen) on neuronal loss seem to be mediated by microglia activation and excessive ROS generation [161], as well as the effects of fibrin(ogen) on astrocytes [162]. At sufficient concentrations, thrombin converts fibrinogen to fibrin, leading to fibrin monomers that are classically associated with blood clots, with proteases, such as tissue plasminogen factor (tPA) converting plasminogen to plasmin [162]. This heightened platelet activation, coupled with increased BBB permeability, contributes to MS symptomatology, including neuronal degeneration and cognitive loss, and is driven at least in part by the impact of coagulation factors on glia activity [162]. Therefore, a means is provided by which variations in platelet activity may be associated with MS [163]. As butyrate suppresses platelet activation, gut dysbiosis will contribute to MS pathophysiology via increased platelet activation and the crossing of coagulation factors over the compromised BBB (see Figure 2).

Increased cholesterol and fat intake that drive gut dysbiosis are associated with raised choline levels, thereby increasing trimethylamine *N*-oxide (TMAO) and TMAO-induced hyperactivated platelets, which contributes to a heightened risk of a blood clot and thrombosis [164]. MS patients have an increased risk of stroke and myocardial infarction in association with altered platelet functioning and pro-thrombotic activity [165]. Platelets contribute to the initial central inflammatory response via elevated levels of IL-1α and interactions with leukocytes [166], whilst platelet depletion ameliorates EAE symptomatology [163]. Classically, activated platelets are associated with vessel inflammatory response due to their ability to adhere to inflamed endothelial cells or their components, as well as via their aggregation with, and activation of, leukocytes. Consequently, any impact of gut dysbiosis on platelet activation is likely to have implications for the pathoetiology and comorbidities of MS.

Gut dysbiosis and increased gut permeability can also heighten TLR activation, which alters the synthesis of the proadhesive hepatic factor von Willebrand factor, thereby modulating the coagulation pathways [167]. Of note, HDAC inhibition can decrease the levels and activity of platelets in preclinical models [168], suggesting that gut dysbiosis associated with reduced butyrate levels may act to potentiate the role of platelets and coagulation factors in MS. Also, HDAC inhibitors, such as butyrate, increase tPA levels, thereby increasing the conversion of plasminogen to plasmin, which breaks down fibrin [169]. This highlights the potential roles of gut-microbiome-derived butyrate in the regulation of increased platelet and coagulation factors in MS, with relevance to core central processes and wider comorbidities in MS pathophysiology, which requires investigation in MS patients.

Of note, platelet aSMase is a major driver of platelet activation and the formation of thrombin [170], suggesting that alterations in the regulation of aSMase/ceramide in MS may also be a core feature in platelets. Higher levels of activated platelets may be primarily associated with SPMS, with SPMS more strongly associated with strokes and myocardial infarction [171]. As data show melatonin to suppress platelet activation [172] and levels [173], the decreased levels of melatonin in MS, in association with lower butyrate levels, will potentiate, if not drive, the role of activated platelets in MS pathophysiology. As with the increased levels of stress and depression in MS [128,174], it would seem that the biological underpinnings of an increased risk of stroke and myocardial infarction may be an integral part of MS pathophysiology, rather than an independent comorbidity. Alterations in the gut microbiome and the regulation of mitochondrial function are inter-related core hubs that connect such variations in physiological functioning across body systems (see Figure 2).

### 3.2. microRNAs and mRNA Binding Proteins

A plethora of recent work shows an important role for microRNAs and mRNA binding/stabilizing proteins (RBPs) in the coordination of cellular responses, including in driving inflammatory responses [175,176]. Controlling the decay of mRNAs that contain adenine-uridine rich elements (AREs) is a common and important aspect of post-transcriptional regulation, including of inflammatory genes. The two most widely investigated RBPs are human antigen R (HuR), which stabilizes ARE-containing mRNA, and tristetraprolin (TTP), which shortens the half-life of ARE-containing mRNA. Data in astrocytes show these RBPs to significantly regulate a wide array of inflammatory inducers, including LPS/TLR2–4, thrombin, and ATP [175]. HuR can also modulate mitochondrial morphology and function [177]. RBPs are important regulators of inflammation, as indicated by HuR inhibition preventing the proinflammatory effects of TNF-α [178].

Gut microbiome-derived butyrate regulates both HuR and TTP in a way that dampens immune inflammatory responses [179]. This requires further investigation in central and systemic cells. However, it does highlight that butyrate has an impact not only by transcriptional regulation, but also by post-transcriptional regulation via RBPs, with consequences that include mitochondria functional and structural regulation.

Likewise, miRNAs are important regulatory coordinators of inflammatory processes and mitochondrial function as shown in pediatric MS [180], at least in part via the regulation of the mitochondrial melatonergic pathway [9]. Given the proposed important role of increased gut permeability in the etiology and course of MS, it is notable that miRNAs can significantly regulate gut permeability and physiological response [181], with the effects of butyrate at least partly mediated via miRNA regulation [182]. A number of studies have highlighted miRNA alterations in MS [183], with the manipulation of gut microbiome-derived butyrate, including by diet and nutraceuticals [184], proposed to mediate benefits, at least in part via the regulation of miRNA-coordinated immune–inflammatory responses. However, miRNAs are an integral aspect of the responses of all the cells and systems showing alteration in MS [183], and the effects of butyrate on such dynamic processes requires investigation. In support of a role for miRNA regulation, the efficacy of fingolimod in MS is at least partly dependent upon miRNA responses [185].

## 4. Future Research

The integrated data in the above model indicate a number of future research directions:(1)Does the modulation of OPCs’ mitochondria by butyrate also act to regulate their remyelinating capacity as well as the putative OPC association with, and disruption of, the BBB?(2)Caffeine increases levels of orexin neuron activity [186], and may decrease the risk, and attenuate the course, of MS [187], including in the EAE model [188]. Are caffeine benefits via orexin neuronal activity, and thereby on daytime mitochondrial function? As caffeine increases peroxisome proliferator-activated receptor gamma coactivator 1-alpha (PGC-1α) and mitochondrial biogenesis in isolated cells, it is likely to also have direct effects on wider mitochondrial functioning [189]. PGC-1α is widely regarded as the mitochondria master regulator;(3)What is the role of the mitochondrial melatonergic pathways within orexin neurons of the lateral hypothalamus/perifornical area?(4)Does butyrate modulate the reactive threshold and activation duration of immune cells via impacts on mitochondrial function [190], including via its activation of the mitochondria melatonergic pathways [28], mediated via PDC disinhibition and increased acetyl-CoA?(5)Given preclinical data indicating greater effects of stress in the inhibition of male orexin neurons [126], is this relevant to sex differences in MS, including increased rates of males shifting to SPMS?(6)As pineal gland-derived melatonin is a significant modulator of the effects of chronic stress in rodents [191], how do variations in pineal melatonin modulate the effects of chronic stress on orexin neuronal structure and function, including across genders?(7)How does the relative loss of orexin activity under chronic stress modulate daytime mitochondrial function in other cells, such as increased glycolysis, and is this reversed by pineal gland derived melatonin, as is proposed in breast cancer cells [41,42]?(8)Is the ceramide inhibition of cellular 14–3-3 levels relevant to mitochondrial 14-3-3 levels and thereby to mitochondrial AANAT stabilization and melatonergic pathway activity?(9)Are ceramide effects via miRNAs associated with 14-3-3 suppression, including by miR-451, miR-375, and miR-7, reviewed in [192]?(10)Is pineal gland NAS/melatonin ratio relevant to MS pathophysiology?(11)Is the increased NAS in SPMS associated with p75NTR activation, leading to increased ceramide synthesis?(12)Are the benefits of exogenous orexin in MS mediated via orexin’s suppression of gut dysbiosis/permeability, with orexin thereby lowering circulating LPS, exosomal HMGB1, and associated brain microglia activation [193]? This would suggest gut, rather than central, effects of exogenous orexin;(13)How relevant are the circadian rhythms of pineal gland-derived melatonin and hypothalamic orexin to the circadian regulation of ceramide [155]?(14)Astrocyte network connectivity, via connexin-43 gap junctions, is important to circadian regulation by orexin [194]. In the absence of a functional lateral hypothalamic astrocyte network and energy provision, animals show excessive sleepiness and difficulty in maintaining alert states [194]. Because reactive astrocytes usually disengage from the astrocyte network, does this indicate that ONOO^-^ or LPS/HMGB1 activation in the lateral hypothalamus may contribute to a decreased synchronization of the astrocyte network and thereby to the suppression of orexin?(15)As well as AhR-induced CYP1b1, a number of other factors can drive the ‘backward’ conversion of melatonin to NAS, including CYP2C19, high ATP levels, mGluR5 activation, and O-demethylation, see [31]. What are the role of these factors in cellular changes in MS?

## 5. Treatment Implications


(1)From the role of the gut microbiome in MS, a number of proposed treatments emerge, including the use of probiotics, fecal microbiota transplantation, bile acid supplementation, and gut barrier enhancers. Sodium butyrate may have more immediate utility. It not only immediately increases the butyrate availability, but also encourages the growth of butyrate-producing gut bacteria. This would have a number of impacts on MS pathophysiology, including the inhibition of ceramide and optimization of mitochondrial function;(2)A number of studies have highlighted the beneficial effects of melatonin in MS patients, with melatonin significantly decreasing the levels of pro-inflammatory cytokines and NO products [195]. However, the effects of melatonin may be enhanced by the adjunctive use of sodium butyrate, which, via the processes outlined above, such as ceramide inhibition and decreasing CYP1b1, would better optimize the effects of melatonin;(3)As noted throughout, fingolimod has many effects on MS pathophysiology that are replicated by the actions of butyrate. Preclinical data show that sodium butyrate facilitates recovery of the BBB following trauma-induced increases in BBB permeability [196]. As BBB permeability has long been associated with MS etiology and relapse, including by permitting the effects of thrombin and fibrin on MS pathophysiology, it requires clinical investigation as to the utility of sodium butyrate as an adjunctive to fingolimod treatment, including as to whether there are additive or synergistic effects in their overlapping modes of efficacy;(4)A combination of daytime orexin and night-time melatonin administration would target the putative loss of the circadian regulation of mitochondrial function and the resetting of mitochondrial oxidative phosphorylation. This will be important to investigate in MS preclinical models, including as to the impact of both treatments on the mitochondrial melatonergic pathway;(5)As urban air quality is a risk factor for pediatric MS [197], with many air pollutants acting via the AhR, and therefore increasing CYP1b1 and the NAS/melatonin ratio, targeting air pollution may be an important preventative strategy. This may be a particular target in SPMS, given its association with increased NAS [156].


## 6. Conclusions

The gut microbiome is intimately involved in the regulation of MS pathophysiology, including via its inhibition of aSMase/ceramides, platelet activation, and the optimization of orexin and melatonin’s circadian modulation of mitochondrial function. The effects of orexin, melatonin, and possibly butyrate, on mitochondrial function seem mediated by their induction of the mitochondrial melatonergic pathway, including via PDC disinhibition and acetyl-CoA upregulation. Such regulation of mitochondria physiology has consequences not only for oligodendrocytes, but also astrocytes, microglia, and systemic immune cell activity. Such a model provides future research directions as well as immediate treatment implications, including the utilization of butyrate and melatonin in the management of MS to prevent disease progression.

## Figures and Tables

**Figure 1 ijms-20-05500-f001:**
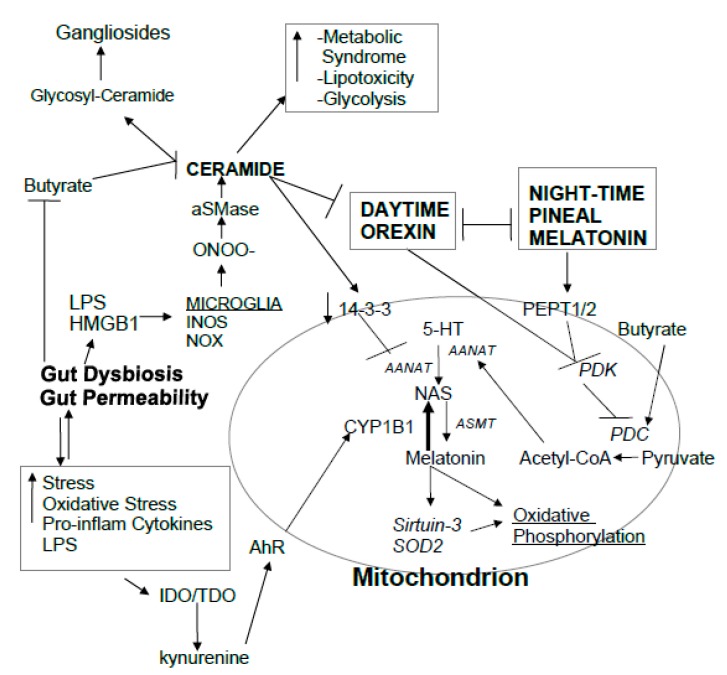
Gut dysbiosis/permeability increase circulating LPS and exosomal HMGB1, leading to TLR activation in microglia and the induction of iNOS and superoxide. This leads to the formation of ONOO^-^, which is a major inducer of aSMase and ceramide. Ceramide has negative impacts on mitochondria functioning, both directly and via the inhibition of orexin and melatonin. Orexin, melatonin, and butyrate disinhibit PDC, leading to an increased conversion of pyruvate to acetyl-CoA, which is the necessary co-substrate for AANAT and the initiation of the mitochondria melatonergic pathway. Increased mitochondria melatonin promotes SOD2, sirtuin-3, and oxidative phosphorylation. By decreasing cellular 14–3-3, ceramide may also inhibit the stabilization of AANAT, thereby preventing the initiation of the melatonergic pathway. Ceramide also increases metabolic syndrome, lipid dysregulation, and gluconeogenesis, all of which are more evident in MS. Gut dysbiosis/permeability have reciprocal interactions with stress, cytokines, and oxidative stress, which can all increase IDO and TDO, leading to kynurenine, which activates the AhR and increases CYP1b1, leading to the backward conversion of melatonin to NAS. Gut dysbiosis/permeability lowers butyrate levels, thereby decreasing butyrate’s suppression of ceramide. Butyrate drives the conversion of ceramide to glucosylceramide and the MS-protective gangliosides. The decrease in butyrate attenuates its inhibition of glia and immune cell reactivity, likely mediated via butyrate regulation of the mitochondrial melatonergic pathway. Abbreviations: AANAT: aralkylamine N-acetyltransferase; AhR: aryl hydrocarbon receptor; aSMase: acid sphingomyelinase; ASMT: N-acetylserotonin O-methyltransferase; CYP: cytochrome P450; HMGB: high-mobility group box; IDO: indoleamine 2,3-dioxygenase; iNOS: inducible nitric oxide; synthase; LPS: lipopolysaccharide; NAS: N-acetylserotonin; NOX/NADPH oxidase: nicotinamide adenine dinucleotide phosphate oxidase; ONOO-: peroxynitrite; PDC: pyruvate dehydrogenase complex; PDK: pyruvate dehydrogenase kinase; PEPT: peptide transporter; SOD2: manganese superoxide dismutase; TDO: tryptophan 2,3-dioxygenase; TLR: toll-like receptor.

**Figure 2 ijms-20-05500-f002:**
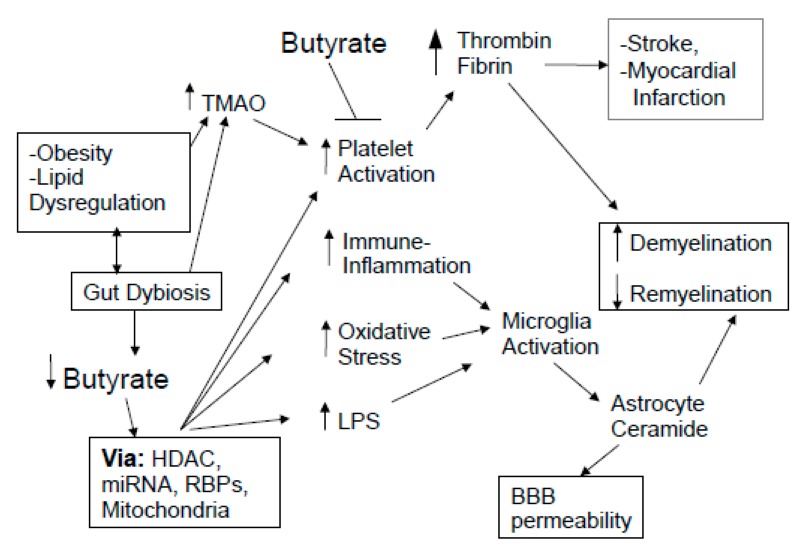
Gut dysbiosis and decreased butyrate contribute to increase immune-inflammation, oxidative stress, LPS, and platelet activation, which act through microglia and astrocyte ceramide, to drive demyelination and suppress remyelination, coupled with an increase in BBB permeability. The two-way interactions of obesity/lipid dysregulation with gut dysbiosis contribute to heightened levels of TMAO. Both TMAO and the consequences of decreased butyrate lead to platelet activation and thereby to an increase in central thrombin and fibrin crossing over a compromised BBB. Raised levels of thrombin/fibrin contribute to the heightened risk of stroke and myocardial infarction in MS, as well as driving demyelination and suppressing remyelination. Abbreviations: BBB: blood-brain barrier; HDAC: histone deacetylase; LPS: lipopolysaccharide; NAS: *N*-acetylserotonin; miRNA: microRNA; RBP; mRNA binding protein; TMAO: trimethylamine N-oxide.

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
