# Peer review of "Multiple Sclerosis: Melatonin, Orexin, and Ceramide Interact with Platelet Activation Coagulation Factors and Gut-Microbiome-Derived Butyrate in the Circadian Dysregulation of Mitochondria in Glia and Immune Cells"

_ijms, 2019, doi:10.3390/ijms20215500_

Round 1

Reviewer 1 Report

This is an interesting review about the role of melatonin and orexin in the circadian regulation of mitochondria functioning and how this may be intimately linked to alterations in the gut microbiome and, therefore, related with multiple sclerosis.

The paper is well written, creates a nice summary and highlights the most important aspects of the role of gut microbiome in the regulation of multiple sclerosis pathophysiology including the inhibition of orexin and melatonin's circadian modulation of mitochondrial functioning.

The two figures of the manuscript are well-constructed and detailed.

The conclusions of the manuscript emphasize the role of gut microbiome in multiple sclerosis pathophysiology and the authors propose the utilization of butyrate and melatonin in the management of multiple sclerosis to prevent disease progression.

Author Response

This is an interesting review about the role of melatonin and orexin in the circadian regulation of mitochondria functioning and how this may be intimately linked to alterations in the gut microbiome and, therefore, related with multiple sclerosis.

The paper is well written, creates a nice summary and highlights the most important aspects of the role of gut microbiome in the regulation of multiple sclerosis pathophysiology including the inhibition of orexin and melatonin's circadian modulation of mitochondrial functioning.

The two figures of the manuscript are well-constructed and detailed.

The conclusions of the manuscript emphasize the role of gut microbiome in multiple sclerosis pathophysiology and the authors propose the utilization of butyrate and melatonin in the management of multiple sclerosis to prevent disease progression.

Response: We would like to take this opportunity to thank the reviewer.

Reviewer 2 Report

General comment:

The commentary is interesting. However, to my mind is long - the multiple aspects which authors deal with could be divided into the parts. The numerous complicated mechanism and sometimes, to my mind, loss of the logical structure, make difficult to understand the hypotheses proposed by authors.

Abstract: Recent data highlights – or highlight?

Please ensure that the affiliations of authors has been written properly. To my opinion, the dot after affiliations is not required.

After the first proposed mechanism in the abstract is dot. The next sentence should be started capitalized.

Abstract, Paragraph 25: CoA – it is an abbreviation, the long-form is required.

Page 2, Paragraph 15 – including mitochondria or disruption of mitochnodria/mitochondrial discfunction?

Figure 1: To my mind, in the pdf version of the manuscript the figure is not completely – the piece of them is cut off (Figure 2 also). Understanding and interpretation are impossible.

Page 6, paragraph 13: t cells – the name of cells should be capitalized.

Page 7 paragraph 33: Human Ig(MS) – please looking reference is proper – The reference no 65 is the article about human mAbs.

Page 8 paragraph 27; the format of citation is improper - please change it according to guidelines

Page 9; Paragraph 6: orexin antagonist has been proposed as ameliorating the severity of depression symptoms rather than the drug for treatment of depression

Page 14: Figure 2 required correction

Page 16, paragraph 13: The font is different from that on the other pages

The text has some grammar incorrect phrases f.e. "The role for circadian dysregulation" or "showing that shift-workers have an increased risk of MS [7], with a preclinical model of shift-work" and some nouns are without „a/an” or „the”

Author Response

The commentary is interesting. However, to my mind is long - the multiple aspects which authors deal with could be divided into the parts. The numerous complicated mechanism and sometimes, to my mind, loss of the logical structure, make difficult to understand the hypotheses proposed by authors.

Response to Reviewer: Thank you for highlighting this. The manuscript aims to integrate previously diverse data contibuting to its length and 'data density'. We have made numerous language and sentence structure changes to aid comprehension by the reader, e.g. breaking long sentences into 2 or 3 shorter sentences. 

-Abstract: Recent data highlights – or highlight?

Response to Reviewer: Thank you. Highlights has now been replaced by highlight.

-Please ensure that the affiliations of authors has been written properly. To my opinion, the dot after affiliations is not required.

Response to Reviewer: Thank you. Dots after affiliations have been removed.

-After the first proposed mechanism in the abstract is dot. The next sentence should be started capitalized.

Response to Reviewer: The next sentence has now been capitalized, as suggested.

-Abstract, Paragraph 25: CoA – it is an abbreviation, the long-form is required.

Response to Reviewer: The full term for 'CoA' is now given at first mention.

-Page 2, Paragraph 15 – including mitochondria or disruption of mitochnodria/mitochondrial discfunction?

Response to Reviewer: This has now been corrected. Thank you.

Figure 1: To my mind, in the pdf version of the manuscript the figure is not completely – the piece of them is cut off (Figure 2 also). Understanding and interpretation are impossible.

Response to Reviewer: Thank you for highlighting this. Presumably this has been a problem in transferring figures from doc to pdf. Smaller figures have been submitted to the journal in case this cannot be readily changed.

Page 6, paragraph 13: t cells – the name of cells should be capitalized.

Response to Reviewer: This has been changed as proposed.

Page 7 paragraph 33: Human Ig(MS) – please looking reference is proper – The reference no 65 is the article about human mAbs.

Response to Reviewer: This sentence has now been changed to read:

 “Human monoclonal antibodies and human immunoglobulin (Ig)Ms can promote remyelination and significantly raise the levels of myelinated axons in MS preclinical models ”   Reference is appropriate as compares to Ig responses.

Page 8 paragraph 27; the format of citation is improper - please change it according to guidelines

Response to Reviewer: Apologies. The named citation should have been removed.

Page 9; Paragraph 6: orexin antagonist has been proposed as ameliorating the severity of depression symptoms rather than the drug for treatment of depression

Response to Reviewer: Thank you for highlighting this. The text has now been changed to read: “Orexin antagonists can also attenuate depression severity”

Page 14: Figure 2 required correction

Response to Reviewer: A smaller figure has now been submitted to the journal.

Page 16, paragraph 13: The font is different from that on the other pages

Response to Reviewer: This must have occurred in transfer to journal formatting, and has now changed to fit the font of the rest of the manuscript.

The text has some grammar incorrect phrases f.e. "The role for circadian dysregulation" or "showing that shift-workers have an increased risk of MS [7], with a preclinical model of shift-work" and some nouns are without „a/an” or „the”

Response to Reviewer: Grammar has now been improved, as suggested.

Reviewer 3 Report

This paper is an extensive (eventually adequately contracted in the manuscript) comprehensive review discussing several elements, although not necessarily directly related, on the multifactorial molecular complexity of MS etiology. The paper is an ambitious attempt to cover extensive territory of diverse areas of research in MS but the themes will be of interest to your reading audience. The paper is well-written and designed. It provides pointers for future research with a therapeutic potential. It is written by known experts in the field. 

Author Response

This paper is an extensive (eventually adequately contracted in the manuscript) comprehensive review discussing several elements, although not necessarily directly related, on the multifactorial molecular complexity of MS etiology. The paper is an ambitious attempt to cover extensive territory of diverse areas of research in MS but the themes will be of interest to your reading audience. The paper is well-written and designed. It provides pointers for future research with a therapeutic potential. It is written by known experts in the field.

Response: We would like to take this opportunity to thank the reviewer.

Reviewer 4 Report

The review covers a very broad range of questions connected with multiple sclerosis (MS). Authors make an afford to integrate the diversity of data on MS, providing the evidence that gut microbiome play a key role in the pathophysiological conditions of MS. The idea of the review is interesting and novel. However, I have some comments regarding the organization of the manuscript.

The abstract does not provide a clear idea of the message the authors want to say. Restructure of abstract and providing more clear links between existing data could improve it. On the other hand, the abstract is overload with details and abbreviations, which makes it hard to follow the idea. That should be balanced. It was really hard to follow the link between MS, gut dysbiosis and permeability in introduction with mitochondria dysfunction in microglia and immune cells, as it seems that authors jump from one subject to another. Attempt to overview all the diversity of data about MS pathology leads to the reader’s confusion. One or two major logical lines of reasoning with some additional details (if needed) could lighten the reading. Page 2 line 22 “Microglia activation leads to ONOO- and tumor necrosis factor…”. It seems that some words missed in this part of the sentence.   A link between melatonin and mitochondrial functioning is provided on the 4th page (Pineal Gland and Mitochondria Melatonin), but there is the only citation. Please state clear facts that are hypothesized and showed previously and published. It is needed to improve the organization of signatures in figures. There are overlapping of text boxes in the schemes and some parts of the text are not printed out. Please pay attention to figures. There are a lot of complex sentences through all over the draft that are could be separated into several simple sentences. That will simplify the understanding of the context.

The overall impression is that the draft is overload with details in some parts that makes it hard to follow the major authors' idea. Reorganization of key paragraphs (abstract, introduction and the initial part of the “Integrating MS Pathophysiology” paragraph) could make it more rider-friendly.

Author Response

The review covers a very broad range of questions connected with multiple sclerosis (MS). Authors make an afford to integrate the diversity of data on MS, providing the evidence that gut microbiome play a key role in the pathophysiological conditions of MS. The idea of the review is interesting and novel. However, I have some comments regarding the organization of the manuscript.

Response to Reviewer: Thank you for these positive comments. 

The abstract does not provide a clear idea of the message the authors want to say. Restructure of abstract and providing more clear links between existing data could improve it. On the other hand, the abstract is overload with details and abbreviations, which makes it hard to follow the idea. That should be balanced.

Response to Reviewer: The data listed in the abstract have now been better linked to make it more comprehensible for the reader, e.g. with the addition of: “ The loss of mitochnodria melatonin, coupled to increased NAS has implication for altered mitochondrial function in many cell types that is relevant to MS pathophysiology. NAS is increased in secondary progressive MS, indicating a role for changes in the mitochondria melatonergic pathway in the progression of MS symptomatology”

It was really hard to follow the link between MS, gut dysbiosis and permeability in introduction with mitochondria dysfunction in microglia and immune cells, as it seems that authors jump from one subject to another. Attempt to overview all the diversity of data about MS pathology leads to the reader’s confusion. One or two major logical lines of reasoning with some additional details (if needed) could lighten the reading.

Response to Reviewer: The manuscript attempts to integrate wider bodies of previously disparate data, which can make for heavy reading. In order to clarify the introduction and better set a framework for the rest of the manuscript, the following has been added to the introduction:

 “As such, gut dysregulation modulates MS pathophysiology via a number of routes: 1)  LPS ultimately activates ceramde, which increases apoptotic susceptibility via detrimental impact on mitochondrial function; 2) ceramde and associated inflammatory cytokines suppress the wake promoting and sleep promoting effects of orexin and melatonin, respectively; 3) the suppression of orexin and melatonin disrupts the circadian rhythm, including from the loss of the mitochondria optimizing effects of orexin and melatonin; 4) the attenuation of gut butyrate production contribute to suboptimal mitochondria functioning, which increases apoptotic susceptibility as well as the reactivity of immune cells, glia and platelets, contributing to a wider pro-inflammatory mileau.”

It should also be noted that the lack of appropriate transfer of the figures to the pdf has probably contributed to the difficulty for the reader to link the broad bodies of data covered in this manuscript. Hopefully, the shrunken figures supplied to the journal will aid reader comprehension.

Page 2 line 22 “Microglia activation leads to ONOO- and tumor necrosis factor…”. It seems that some words missed in this part of the sentence.

Response to Reviewer: This sentence has now been changed to read: “Microglia activation increases the production and release of tumor necrosis factor (TNF)-α and peroxynitrite (ONOO-), with the latter elevating levels of astrocyte acidic sphingomyelinase (aSMase), in turn increasing ceramide release, including within exosomes.”

A link between melatonin and mitochondrial functioning is provided on the 4th page (Pineal Gland and Mitochondria Melatonin), but there is the only citation.

Response to Reviewer: Thank you for highlighting this. A further two references have now been provided in this section.

Please state clear facts that are hypothesized and showed previously and published.

Response to Reviewer: All data is based on published studies, apart from novel proposed connections.

It is needed to improve the organization of signatures in figures. There are overlapping of text boxes in the schemes and some parts of the text are not printed out. Please pay attention to figures.

Response to Reviewer: Thank you for highlighting this. Presumably this has been a problem in transferring figures from doc to pdf. Smaller figure have been submitted to the journal in case this cannot be readily changed and should aid reader comprehension. 

There are a lot of complex sentences through all over the draft that are could be separated into several simple sentences. That will simplify the understanding of the context.

Response to Reviewer: Thank you for highlighting this. A number of sentences have now been broken into shorter sentences, as suggested.

The overall impression is that the draft is overload with details in some parts that makes it hard to follow the major authors' idea. Reorganization of key paragraphs (abstract, introduction and the initial part of the “Integrating MS Pathophysiology” paragraph) could make it more rider-friendly.

Response to Reviewer:  Hopefully, the changes in the abstract, introduction and intial integration sections now set a better framework for the reader to integrate the necessary broad bodies of data covered in this manuscript. Being able to appropriately view the summary figures should also greatly aid the reader to integrate the wide bodies of data forming this submission.

Round 2

Reviewer 2 Report

The Editorial Board should pay attention to the font in the text (f.e. References has two different types). Probably, as Authors have noticed, this is the result in transfer to journal formatting).